# An Alternative Electro-Mechanical Finite Formulation for Functionally Graded Graphene-Reinforced Composite Beams with Macro-Fiber Composite Actuator

**DOI:** 10.3390/ma14247802

**Published:** 2021-12-16

**Authors:** Yu Fu, Xingzhong Tang, Qilin Jin, Zhen Wu

**Affiliations:** 1Advanced Rotorcraft Department, Chinese Aeronautical Establishment, Beijing 100101, China; tangxingzhong@cae.ac.cn; 2School of Aeronautics, Northwestern Polytechnical University, Xi’an 710072, China; jinqilin0404@nwpu.edu.cn (Q.J.); wuzhenhk@nwpu.edu.cn (Z.W.)

**Keywords:** graphene-reinforced composite, FG-GRC-laminated beam, electro-mechanical coupled-beam model, transverse shear stresses

## Abstract

With its extraordinary physical properties, graphene is regarded as one of the most attractive reinforcements to enhance the mechanical characteristics of composite materials. However, the existing models in the literature might meet severe challenges in the interlaminar-stress prediction of thick, functionally graded, graphene-reinforced-composite (FG-GRC)-laminated beams that have been integrated with piezoelectric macro-fiber-composite (MFC) actuators under electro-mechanical loadings. If the transverse shear deformations cannot be accurately described, then the mechanical performance of the FG-GRC-laminated beams with MFC actuators will be significantly impacted by the electro-mechanical coupling effect and the sudden change of the material characteristics at the interfaces. Therefore, a new electro-mechanical coupled-beam model with only four independent displacement variables is proposed in this paper. Employing the Hu–Washizu (HW) variational principle, the precision of the transverse shear stresses in regard to the electro-mechanical coupling effect can be improved. Moreover, the second-order derivatives of the in-plane displacement parameters have been removed from the transverse-shear-stress components, which can greatly simplify the finite-element implementation. Thus, based on the proposed electro-mechanical coupled model, a simple C^0^-type finite-element formulation is developed for the interlaminar shear-stress analysis of thick FG-GRC-laminated beams with MFC actuators. The 3D elasticity solutions and the results obtained from other models are used to assess the performance of the proposed finite-element formulation. Additionally, comprehensive parametric studies are performed on the influences of the graphene volume fraction, distribution pattern, electro-mechanical loading, boundary conditions, lamination scheme and geometrical parameters of the beams on the deformations and stresses of the FG-GRC-laminated beams with MFC actuators.

## 1. Introduction

Composite structures with sensors and actuators, which are called smart structures, are used in a wide range of engineering applications [1,2,3]. Among the various available smart materials, the piezoelectric material is one of the most commonly used materials that can be used as both actuators and sensors. Macro-fiber composites (MFCs) are a new type of piezoelectric intelligent materials that were developed by NASA. The rectangular cross-section of the piezoelectric fibers in the MFC increases the contact area between the interdigitated electrodes and piezoceramic fibers in order to improve the transfer of electricity [4,5,6]. With a higher electro-mechanical conversion efficiency and better flexibility, the MFC has broad application prospects and research value.

In order to achieve the vibration control and shape simulation of smart structures with MFC patches, Zhang et al. [7] studied the active shape and vibration control of laminated plates with MFCs. Guo et al. [8] analyzed the nonlinear dynamics of MFC piezoelectric plates with graphene skins. Dong et al. [9] proposed an equivalent-force-modeling approach to investigate plate-type structures that were integrated with MFC actuators. Rao et al. [10] carried out the large-deflection electro-mechanical analysis of laminated composite structures that were bonded with MFC actuators under thermo-electro-mechanical loads. Gawryluk et al. [11] studied the dynamic behavior of a composite beam that was rotating at a constant angular velocity and excited by an MFC actuator. In addition, Gawryluk et al. [12] analyzed the problem of vibration reduction in a cantilever beam with an embedded MFC actuator under kinematic excitation. Zhou et al. [13] investigated the aeroelastic stability of curved composite panels with embedded MFC actuators in a supersonic airflow.

Recently, functionally graded composites that are reinforced with carbon nanotubes and graphene have emerged as the new generation of advanced composite materials with greater improvements of physical [14], chemical [15] and mechanical properties [16] compared to composites without reinforcement. Due to the outstanding mechanical properties of graphene, the investigations into the FG-GRC-laminated structures that are integrated with piezoelectric actuators are significant in a wide range of engineering applications. For reliable and safe designs, effective models are required in order to study the mechanical behaviors of smart FG-GRC-laminated structures. Due to the limitations of the existing manufacturing techniques, it is extremely difficult to fabricate graphene-reinforced composites with continuous material-characteristic distributions through the thickness [17]. In addition, the multilayer configuration of nanocomposites composed of stacked layers was introduced. Graphene was uniformly dispersed in each layer, and the graphene volume fraction gradually changed layer by layer. Based on the first-order shear-deformation theory (FSDT), Song et al. [17] studied the free and forced vibration behaviors of FG-GRC plates. Then, investigations into the bending [18], buckling and postbuckling [19], nonlinear free vibration [20], and dynamic instability [21,22] of FG-GRC structures were reported by Yang et al. Kiani et al. [23,24,25,26,27,28] carried out the buckling and postbuckling analysis of FG-GRC structures in a thermal environment. Reddy et al. [29] investigated the free vibration of GRC plates via the finite-element method. Guo et al. [30] proposed an element-free IMLS-Ritz method for the vibration analysis of GRC-laminated plates. Zhang et al. [31] proposed the DSC-regularized Dirac-delta method for the vibration analysis of porous FG-GRC beams that were resting on an elastic foundation under a moving load. Liu et al. [32] investigated the dynamic behaviors of porous FG-GRC plates that were resting on an elastic foundation. By using the spectral-Chebyshev approach, free vibration and buckling responses of porous FG-GRC plates were conducted by Anamagh and Bediz [33]. In a review of the aforementioned works, FSDT was used to analyze the FG-GRC structures. However, the accuracy of FSDT was impacted by the shear correction factor. In order to avoid using the shear correction factor, various higher-order shear-deformation theories (HSDT) with nonlinear shear strains were used for the analysis of the FG-GRC structures. Using Reddy’s higher-order theory, Shen et al. studied the nonlinear bending [34], buckling and postbuckling [35,36], and nonlinear vibration [37,38] of FG-GRC-laminated structures in thermal environments. Wang et al. [39] investigated the free vibration and bending of doubly-curved FG-GRC shallow shells. By applying the sinusoidal shear-deformation theory, Arefi et al. [40] analyzed the free vibration of FG-GRC nanoplates. A statistical analysis of the free vibration of FG-GRC plates was conducted by Pashmforoush [41]. In terms of the refined four-variable model, a NURBS formulation for the bending, buckling and free vibration analysis of multilayer FG-GRC plates was proposed by Thai et al. [42]. The vibro-acoustic analysis of FG-GRC-laminated plates was achieved by Xu and Huang [43]. Based on the three-dimensional, refined higher-order shear-deformation theory, Al-Furjan et al. [44] studied the bending responses of an FG-GRC disk. More studies of the mechanical analysis of FG-GRC structures can be found in the published articles [45,46,47,48,49].

Static, buckling and dynamic analyses of laminated composite plates with MFC actuators have been extensively investigated [8,9,10,11,12,13,50,51,52,53,54]. In addition, piezoelectric materials were used in the smart structures for shape, vibration and buckling control [7,12,55,56,57]. However, the research on the structural behaviors of FG-GRC-laminated plates with piezoelectric actuators is quite limited. Mao and Zhang [58,59] analyzed the linear and nonlinear vibration and buckling behaviors of piezoelectric FG-GRC plates. Using the transformed-differential-quadrature method, Malekzadeha et al. [60] studied the free vibration of eccentric annular FG-GRC plates that were embedded in piezoelectric layers. Nguyena et al. [61] proposed an isogeometric finite-element formulation for the bending and transient analysis of porous FG-GRC plates that were embedded in piezoelectric layers. The free vibration of FG-GRC sandwich plates that were enclosed by piezoelectric layers was investigated by Majidi-Mozafari et al. [62]. The wave-characteristic analysis of cylindrical FG-GRC sandwich structures with piezoelectric surface layers was reported by Li and Han [63,64]. Khayat at al. [65] investigated the uncertainty propagation in the nonlinear dynamic responses of smart, porous cylindrical FG-GRC sandwich shells. Guo et al. [66] investigated the nonlinear dynamic behavior of three-phase composite plates that were made with MFCs in the polymer and with graphene skins.

Owing to the electro-mechanical coupling effect and the discontinuity of the material properties throughout the thickness of the FG-GRC-laminated structures with piezoelectric layers, transverse shear deformations play an important role. In order to precisely predict the transverse shear stresses of FG-GRC-laminated structures with MFC actuators, the compatible conditions of transverse shear stresses at the interfaces must be fulfilled. As can be observed in the above literature, FSDT and HSDT combined with different solution methods are used for the mechanical analysis of FG-GRC structures with piezoelectric layers. However, FSDT and HSDT are unable to satisfy the compatible conditions of transverse shear stresses at the interfaces. As a result, FSDT and HSDT may fail to model the transverse shear stresses of the FG-GRC-laminated plates with MFC actuators. Therefore, there is a genuine requirement to propose an efficient and accurate model with which to study this issue.

From the above literature survey, it was found that the comprehensive interlaminar shear-stress analysis of the thick FG-GRC-laminated beams with MFC actuators are scarce in the open literature. This paper aims to fill this research gap by developing an efficient finite-element formulation based on an attractive electro-mechanically coupled model. The proposed model fulfills the compatible conditions of transverse shear stresses at the interfaces. By using the 3D elasticity equations and the HW variational principle [67,68], the accuracy of the transverse shear stresses in terms of the electro-mechanical coupling effect is improved. Due to the second-order derivatives of the in-plane displacement, variables are removed from the interlaminar shear-stress components, such that the finite-element formulation can be easily developed. Thus, based on the proposed electro-mechanical model, a simple C^0^-type two-node beam element is developed for the interlaminar-stress analysis of thick FG-GRC-laminated beams with MFC actuators. The performance of this finite-element formulation is appraised by comparing it with the 3D elasticity solutions and the results are calculated from the chosen models. Comprehensive parametric studies are conducted in order to explore the influences of distribution pattern, volume fraction, lamination sequence, as well as geometric parameters of the beams on the deformations and stresses of FG-GRC-laminated beams with MFC actuators. The research findings will provide essential information for the reliable design of smart FG-GRC-laminated structures with great potential for engineering applications.

## 2. Basic Formulations

### 2.1. Effective Material Properties of GRC

An FG-GRC-laminated beam integrated with an MFC layer is shown in Figure 1. The length of the beam is *a*. The thicknesses of the FG-GRC-laminated beam and the MFC-actuator layer are *h* and *h*_p_, respectively. Each GRC layer may have a different graphene volume fraction. The graphene reinforcement distribution is functionally graded and of the piece-wise type in terms of thickness. The four distribution patterns of the FG-GRC beam are shown in Figure 2, denoted as UD, FG-V, FG-O and FG-X. Different colors are used to represent the distributions of the volume fraction in terms of thickness. A dark color indicates a large volume fraction, whereas a light color indicates a small volume fraction. First of all, the graphene volume fractions of the FG-V GRC beams are set to [(0.11)^2^/(0.09)^2^/(0.07)^2^/(0.05)^2^ /(0.03)^2^]. In this type of FG-GRC beam, the volume fraction of the top layer is the largest and the volume fraction of the bottom layer is the smallest. Secondly, the graphene volume fractions are arranged as [0.11/0.09/0.07/0.05/0.03]_s_ for a ten-layer beam and expressed as FG-X. Different from the FG-V type, the top and bottom GRC layers have the largest graphene volume fraction and the layers near the mid-surface have the smallest graphene volume fraction. Then, the arrangement of graphene volume fractions [0.03/0.05/0.07/0.09/0.11]_s_ is referred to FG-O. In this type, the volume fraction is maximum at the mid-surface and minimum at the top and bottom layers. Finally, for comparative purposes, the type of UD GRC beams with the same thickness and graphene volume fraction *V_G_* = 0.07 are considered.

Based on the extended Halpin–Tsai model [38], the effective elastic moduli of the GRC layer are given by
(1)$$\begin{array}{l} E_{11}=\eta_{1}\frac{1+2\left(a_{G}/h_{G}\right)\gamma_{11}V_{G}}{1-\gamma_{11}V_{G}}E^{m}\\ E_{22}=\eta_{2}\frac{1+2\left(b_{G}/h_{G}\right)\gamma_{22}V_{G}}{1-\gamma_{22}V_{G}}E^{m}\\ G_{12}=\eta_{3}\frac{1}{1-\gamma_{12}V_{G}}G^{m} \end{array}$$
where
(2)$$\begin{array}{l} \gamma_{11}=\frac{E_{11}^{G}/E^{m}-1}{E_{11}^{G}/E^{m}+2a_{G}/h_{G}}\\ \gamma_{22}=\frac{E_{22}^{G}/E^{m}-1}{E_{22}^{G}/E^{m}+2b_{G}/h_{G}}\\ \gamma_{12}=\frac{G_{12}^{G}/E^{m}-1}{G_{12}^{G}/E^{m}} \end{array}$$
in which the superscripts *G* and *m* denote graphene and matrix, $a_{G}$, $b_{G}$ and $h_{G}$ are the effective length, width and thickness of the graphene sheet, respectively. The volume fractions $V_{G}$ and $V_{m}$ satisfy the condition $V_{G}+V_{m}=1$. The graphene efficiency parameters $\eta_{i}\left(i=1,2,3\right)$ in Equation (1) can be obtained by the MD simulation.

The density and Poisson ratio of the composite media can be expressed as
(3)$$\begin{array}{l} \rho =V_{G}\rho^{G}+V_{m}\rho^{m}\\ v_{12}=V_{G}v_{12}^{G}+V_{m}v^{m} \end{array}$$
where $v_{12}^{G}$ and $v_{12}^{m}$ denote the Poisson ratio of graphene sheet and matrix, respectively.

### 2.2. Displacement Field

The displacement field of the present beam theory is a combination of the first-order shear-deformation theory and the efficient zigzag functions, and is given by
(4)$$\begin{array}{l} u^{k}\left(x,z\right)=u_{0}\left(x\right)+zu_{1}\left(x\right)+\Phi^{k}(z)u_{z}(x)\\ w^{k}\left(x,z\right)=w_{0}\left(x\right) \end{array}$$
where superscript *k* represents the *k*th ply of the laminated beam, $u^{k}$ and $w^{k}$ denote the assumptions of in-plane displacement and transverse displacement for the *k*th ply of the beam, $u_{0}$ is the in-plane axial displacement, $u_{1}$ is the bending rotation, $u_{z}$ is the amplitude function and $w_{0}$ is the transverse displacement.

The zigzag function $\Phi^{k}(z)$ in Equation (4) can be explicitly written as
(5)$$\Phi^{k}(z)=\left(1+z^{2}\right)\left[\frac{1}{2}(1-\zeta_{k}){\bar{u}}^{k}+\frac{1}{2}(1+\zeta_{k}){\bar{u}}^{k+1}\right]$$
where ${\bar{u}}^{k}$ are the interfacial axial displacements. $\zeta_{k}\in [-1,1]$, $\zeta_{k}=a_{k}z-b_{k}$, $a_{k}=2/(z_{k+1}-z_{k})$, $b_{k}=(z_{k+1}+z_{k})/(z_{k+1}-z_{k})$, and $z_{k}$ is defined in Figure 1.

In terms of the displacement field given by Equation (4), the transverse shear strain is expressed as
(6)$$\gamma_{xz}^{k}=\eta_{x}+[1+\frac{\partial \Phi^{k}(z)}{\partial z}]u_{z}$$
where
(7)$$\eta_{x}=\frac{\partial w_{0}}{\partial x}+u_{1}-u_{z}$$

The transverse shear stress for the *k*th ply is expressed as
(8)$$\tau_{xz}^{k}=Q_{44}^{k}\eta_{x}+Q_{44}^{k}[1+\frac{\partial \Phi^{k}(z)}{\partial z}]u_{z}=Q_{44}^{k}\eta_{x}+{\bar{\tau }}_{xz}^{k}$$
in which $Q_{44}^{k}$ represent the shear modulus of the *k*th ply

In Equation (8), it is found that the transverse shear stress contains two parts: the continuity conditions are applied only on the zigzag-dependent part, ${\bar{\tau }}_{xz}^{k}$, that is, on the transverse shear stresses obtained by removing the shear measures $\eta_{x}$. The continuity conditions are expressed as
(9)$${\bar{\tau }}_{xz}^{k}(x,z_{k+1})={\bar{\tau }}_{xz}^{k+1}(x,z_{k+1})$$

Equation (9) provides *N-*1 conditions for the interfacial axial displacements. The other two conditions can be obtained from the zero-value conditions on the outer plate surfaces, which are given by
(10)$${\bar{\tau }}_{xz}^{b}(x,-\frac{h}{2})=0,\;{\bar{\tau }}_{xz}^{t}(x,\frac{h}{2})=0$$
where *t* and *b* are used to denote the bottom and top layers of the beam.

According to the Equations (9) and (10), the ${\bar{u}}^{k}$ given by Equation (5) are obtained. Based on the assumed displacement field, the strains are shown as
(11)$${\left\{\begin{array}{c} \varepsilon_{x}\\ \gamma_{xz} \end{array}\right\}}^{k}=\left[\begin{array}{cc} B_{IX} & B_{TXZ} \end{array}\right]\left\{\begin{array}{c} D_{IX}\\ D_{TXZ} \end{array}\right\}$$
where
(12)$$\begin{array}{l} B_{IX}=\left\{\begin{array}{ccc} 1 & z & \Phi^{k} \end{array}\right\}\\ B_{TXZ}=\left\{\begin{array}{ccc} 1 & \frac{\partial \Phi^{k}}{\partial z} & 1 \end{array}\right\} \end{array}$$
(13)$$\begin{array}{l} D_{IX}={\left\{\begin{array}{ccc} \frac{\partial u_{0}}{\partial x} & \frac{\partial u_{1}}{\partial x} & \frac{\partial u_{z}}{\partial x} \end{array}\right\}}^{T}\\ D_{TXZ}={\left\{\begin{array}{ccc} u_{1} & u_{z} & \frac{\partial w_{0}}{\partial x} \end{array}\right\}}^{T} \end{array}$$

The constitutive relations for the *k*th ply of a cross-ply FG-GRC-laminated beam are given by
(14)$${\left\{\begin{array}{c} \sigma_{x}\\ \tau_{xz} \end{array}\right\}}^{k}={\left[\begin{array}{cc} Q_{11} & 0\\ 0 & Q_{44} \end{array}\right]}^{k}{\left\{\begin{array}{c} \varepsilon_{x}\\ \gamma_{xz} \end{array}\right\}}^{k}$$
where $Q_{ij}^{k}$ are the transformed elastic stiffness coefficients of the *k*th ply.

For the piezoelectric layer, the stress components are given by [69]
(15)$${\left\{\begin{array}{c} \sigma_{x}\\ \tau_{xz} \end{array}\right\}}^{k}={\left[\begin{array}{cc} Q_{11} & 0\\ 0 & Q_{44} \end{array}\right]}^{k}{\left\{\begin{array}{c} \varepsilon_{x}\\ \gamma_{xz} \end{array}\right\}}^{k}-{\left[\begin{array}{cc} 0 & e_{31}\\ e_{15} & 0 \end{array}\right]}^{k}{\left\{\begin{array}{c} E_{x}\\ E_{z} \end{array}\right\}}^{k}$$
where $e_{ij}^{k}$ are the transformed piezoelectric modulus of the *k*th ply. The electric field *E* can be obtained by the following equation
(16)$${\left\{\begin{array}{c} E_{x}\\ E_{z} \end{array}\right\}}^{k}=-{\left\{\begin{array}{c} \frac{\partial \xi }{\partial x}\\ \frac{\partial \xi }{\partial z} \end{array}\right\}}^{k}$$
where ξ denotes the electrostatic potential, which is assumed as
(17)$$\xi \left(x,y,z\right)=V_{t}\left(\frac{2z-h}{2h_{p}}\right)\phi \left(x\right)$$
in which $\phi \left(x\right)$ represents the axial distribution of the applied electrostatic potential, *V_t_* and *h_p_* denote the applied electric voltage and the thickness of the MFC layer, respectively.

### 2.3. Improved Transverse Shear Stresses Considering Electro-Mechanical Coupling Effect

Owning to the electro-mechanical coupling effect and the sudden change of material characteristics at the interfaces of the FG-GRC-laminated beams with piezoelectric layers, the transverse shear stresses that are calculated directly from the constitutive equations have low accuracy. To achieve an improved through-the-thickness variation of transverse shear stresses in terms of the electro-mechanical coupling effect, the 3D equilibrium equations and HW variational principle are used. The key feature of this formulation is that the assumed transverse shear stresses have a superior accuracy.

For the laminated beam, the 3D equilibrium equation can be reduced to the 2D equilibrium equation. Disregarding the influence of the body force, the 2D equilibrium equation is expressed as
(18)$$\frac{\partial \sigma_{x}^{k}}{\partial x}+\frac{\partial \tau_{xz}^{k}}{\partial z}=0$$

Integrating Equation (18) with respect to the *z*-coordinate and enforcing the condition ${\tilde{\tau }}_{xz}^{k}(-h/2)=0$ yields the following form for the transverse shear stress
(19)$${\tilde{\tau }}_{xz}^{k}(x,z)=-\int_{-h/2}^{z}\frac{\partial \sigma_{x}^{k}}{\partial x}dz$$

Substituting Equations (11)–(17) into Equation (19), yields
(20)$${\tilde{\tau }}_{xz}^{k}(x,z)=-C_{x}(z)U_{x}-C_{\phi }(z)U_{\phi }$$
where
(21)$$C_{x}(z)=\int_{-h/2}^{z}Q_{11}^{k}\left[\begin{array}{ccc} 1 & z & \Phi^{k} \end{array}\right]dz,\;C_{\phi }(z)=\int_{-h/2}^{z}e_{31}^{k}\frac{V_{t}}{h_{p}}dz$$
(22)$$U_{x}=\frac{\partial^{2}}{\partial x^{2}}{\left[\begin{array}{ccc} u_{0} & u_{1} & u_{z} \end{array}\right]}^{T},\;U_{\phi }=\frac{\partial \phi }{\partial x}$$

By applying the condition ${\tilde{\tau }}_{xz}^{k}(h/2+h_{p})=0$, the second-order derivative function $\partial^{2}u_{0}/\partial x^{2}$ is removed from the Equation (20), yielding
(23)$${\tilde{\tau }}_{xz}^{k}(x,z)=F_{x}(z){\bar{U}}_{x}+F_{\phi }(z)U_{\phi }$$
where
(24)$$\begin{array}{l} F_{x}(z)=\int_{-h/2}^{z}Q_{11}^{k}\left[\begin{array}{cc} A_{1} & A_{2} \end{array}\right]\mathrm{dz}\\ F_{\phi }(z)=\int_{-h/2}^{z}\left(Q_{11}^{k}A_{3}-e_{31}^{k}\frac{V_{t}}{h_{p}}\right)\mathrm{dz} \end{array}$$
(25)$${\bar{U}}_{x}=\frac{\partial^{2}}{\partial x^{2}}{\left[\begin{array}{cc} u_{1} & u_{z} \end{array}\right]}^{T}$$
in which
(26)$$A_{1}=\frac{\int_{-h/2}^{h/2}Q_{11}^{k}zdz}{\int_{-h/2}^{h/2}Q_{11}^{k}dz}-z,\;A_{2}=\frac{\int_{-h/2}^{h/2}Q_{11}^{k}\Phi^{k}dz}{\int_{-h/2}^{h/2}Q_{11}^{k}dz}-\Phi^{k},\;A_{3}=\frac{\int_{-h/2}^{h/2}e_{31}^{k}\frac{V_{t}}{h_{p}}dz}{\int_{-h/2}^{h/2}Q_{11}^{k}dz}$$

The assumed transverse shear stress ${\tilde{\tau }}_{xz}^{k}$ in Equation (23) can a priori fulfill the interlaminar continuity conditions at the interfaces and the zero-traction conditions on the outer beam surfaces. The transverse shear stresses contain the second-order derivatives of the in-plane displacement variables, which will lead to difficulties in finite-element modeling. Thus, the HW variational principle [67,68] is employed to eliminate the second-order derivatives of the in-plane displacement parameters. In terms of the HW variational principle, a function for the laminated beams can be stated as
(27)$$\Pi \left(u,\;\varepsilon ,\;\sigma \right)=\int_{\Omega }\left(\frac{1}{2}\varepsilon^{T}Q\varepsilon +\sigma^{T}\left(\varepsilon^{c}\left(u\right)-\varepsilon \right)\right)d\Omega -\Pi^{\mathrm{ext}}\left(u\right)$$
where **u**, **ε** and **σ** are the independent variables that are subjected to variation, and Ω is the region occupied by a whole beam.
(28)$$\varepsilon^{c}={\left\{\begin{array}{cc} \varepsilon_{x}^{k} & \gamma_{xz}^{k} \end{array}\right\}}^{T},\;\varepsilon ={\left\{\begin{array}{cc} \varepsilon_{x}^{k} & {\tilde{\gamma }}_{xz}^{k} \end{array}\right\}}^{T}$$
(29)$$\Pi^{ext}\left(u\right)=\int_{S}qudS$$
in which **q** is a traction vector, and the transverse shear strain ${\tilde{\gamma }}_{xz}^{k}$ corresponding to the assumed transverse shear stress ${\tilde{\tau }}_{xz}^{k}$ is given by
(30)$${\tilde{\gamma }}_{xz}^{k}=G_{x}(z){\bar{U}}_{x}+G_{\phi }(z)U_{\phi }$$
(31)$$G_{x}(z)=\frac{F_{x}(z)}{Q_{44}^{k}},\;G_{\phi }(z)=\frac{F_{\phi }(z)-e_{15}^{k}V_{t}\left(\frac{2z-h}{2h_{p}}\right)}{Q_{44}^{k}}$$

Evaluation of the first variation of the Equation (27) and equating to zero results in
(32)$$\begin{array}{ll} \delta \Pi \left(u,\;\varepsilon ,\;\sigma \right)= & \int_{\Omega }\left(\varepsilon^{T}Q-\sigma^{T}\right)\delta \varepsilon d\Omega +\int_{\Omega }\delta \sigma^{T}\left(\varepsilon^{c}-\varepsilon \right)d\Omega \\ & +\int_{\Omega }\sigma^{T}\delta \varepsilon^{c}\left(u\right)d\Omega -\delta \Pi^{\mathrm{ext}}\left(u\right)=0 \end{array}$$

Applying integration by parts, the following equation is given by
(33)$$\int_{\Omega }\sigma^{T}\delta \varepsilon^{c}\left(u\right)d\Omega -\delta \Pi^{\mathrm{ext}}\left(u\right)=\int_{\Omega }D\sigma^{T}\delta ud\Omega +\int_{S}D_{S}\sigma^{T}\delta udS$$
where **D** and **D**_S_ are the differential operators.

Introducing Equation (33) into Equation (32) yields
(34)$$\begin{array}{ll} \delta \Pi \left(u,\;\varepsilon ,\;\sigma \right)= & \int_{\Omega }\left(\varepsilon^{T}Q-\sigma^{T}\right)\delta \varepsilon d\Omega +\int_{\Omega }\delta \sigma^{T}\left(\varepsilon^{c}-\varepsilon \right)d\Omega \\ & +\int_{\Omega }D\sigma^{T}\delta ud\Omega +\int_{S}D_{S}\sigma^{T}\delta udS \end{array}$$

According to the second integral in Equation (34), the following equation can be given by
(35)$$\int_{\Omega }\delta {\tilde{\tau }}_{xz}^{k}(\gamma_{xz}^{k}-{\tilde{\gamma }}_{xz}^{k})d\Omega =0$$

Introducing Equations (11), (23) and (30) into Equation (35), results in
(36)$$\int_{-h/2}^{h/2+h_{p}}\delta {\bar{U}}_{x}^{T}F_{x}^{T}(B_{TXZ}D_{TXZ}-G_{x}{\bar{U}}_{x}-G_{\phi }U_{\phi })dz=0$$

Using the above equation, the second-order derivatives of the displacement parameters ${\bar{U}}_{x}$ can be rewritten as
(37)$${\bar{U}}_{x}=C_{EXZ}D_{TXZ}-C_{\phi }U_{\phi }$$
where
(38)$$\begin{array}{l} C_{EXZ}={\left[\int_{-h/2}^{h/2+h_{p}}F_{x}^{T}G_{x}dz\right]}^{-1}\left[\int_{-h/2}^{h/2+h_{p}}F_{x}^{T}B_{TXZ}dz\right]\\ C_{\phi }={\left[\int_{-h/2}^{h/2+h_{p}}F_{x}^{T}G_{x}dz\right]}^{-1}\left[\int_{-h/2}^{h/2+h_{p}}F_{x}^{T}G_{\phi }dz\right] \end{array}$$

Introducing Equation (37) into Equation (23) yields the final expression of ${\tilde{\tau }}_{xz}^{k}$
(39)$${\tilde{\tau }}_{xz}^{k}(x,z)=F_{x}C_{EXZ}D_{TXZ}+\left(F_{\phi }-F_{x}C_{\phi }\right)U_{\phi }={Μ}_{1}^{k}u_{1}+{Μ}_{2}^{k}u_{z}+{Μ}_{3}^{k}\frac{\partial w_{0}}{\partial x}+{Μ}_{4}^{k}\frac{\partial \phi }{\partial x}$$

## 3. Finite-Element Formulation

In this section, a two-node beam element is used in order to study the FG-GRC-laminated beams with MFC actuators. Accordingly, the displacement variables for each element can be expressed as
(40)$$\left\{\begin{array}{c} u_{0}\\ u_{1}\\ u_{z}\\ w_{0} \end{array}\right\}=\sum_{i=1}^{2}\left[\begin{array}{cccc} N_{i} & 0 & 0 & 0\\ 0 & N_{i} & 0 & 0\\ 0 & 0 & N_{i} & 0\\ 0 & 0 & 0 & N_{i} \end{array}\right]\left\{\begin{array}{c} u_{0i}\\ u_{1i}\\ u_{zi}\\ w_{0i} \end{array}\right\}$$
where $N_{1}=(1-\xi )/2$, $N_{2}=(1+\xi )/2$; $\xi =(x-x_{c})/l_{e}$, $x_{c}=x_{i}+l_{e}/2$, $l_{e}=x_{i+1}-x_{i}$; $x_{i}$ and $x_{i+1}$ are defined in Figure 3.

By means of Equations (11)–(13), the strain vector is written as
(41)$$\varepsilon^{k}=\partial u^{k}=Bd^{e}$$
in which $u^{k}={\left[\begin{array}{cc} u & w \end{array}\right]}^{T}$, $B=\left[\begin{array}{cc} B_{1} & B_{2} \end{array}\right]$, $d^{e}={\left[\begin{array}{cc} d_{1}^{e} & d_{2}^{e} \end{array}\right]}^{T}$ and $d_{i}^{e}$ are the degrees of freedom at the *i*th node. The degrees of freedom at each node are given by
(42)$$d_{i}^{e}={\left[\begin{array}{cccc} u_{0i} & w_{0i} & u_{1i} & u_{zi} \end{array}\right]}^{T}$$
(43)$$\partial =\left[\begin{array}{cc} \frac{\partial }{\partial x} & 0\\ \frac{\partial }{\partial z} & \frac{\partial }{\partial x} \end{array}\right]$$
(44)$$B_{i}=\left[\begin{array}{cccc} \frac{\partial N_{i}}{\partial x} & 0 & z\frac{\partial N_{i}}{\partial x} & \Phi^{k}\frac{\partial N_{i}}{\partial x}\\ 0 & \frac{\partial N_{i}}{\partial x} & N_{i} & \frac{\partial \Phi^{k}}{\partial z}N_{i} \end{array}\right]$$

According to the Equation (33), the element-stiffness matrix of the proposed beam element can be obtained. Equation (33) within the *i*th element can be rewritten as
(45)$$\int_{x_{i}}^{x_{i+1}}\int_{-h/2}^{h/2+h_{p}}\left(\sigma_{x}^{k}\delta \varepsilon_{x}^{k}+{\tilde{\tau }}_{xz}^{k}\delta \gamma_{xz}^{k}\right)dzdx-\delta W_{i}=0$$
where *W_i_* represents the work of external forces.

Introducing Equations (11)–(17), (39) into Equation (45), the element-stiffness matrix is expressed as
(46)$$K^{e}=\int_{e}B^{T}Q\hat{B}dxdz$$
in which **Q** is the material-constant matrix, the strain matrix **B** is given in Equation (44), and the strain matrix $\hat{B}$ is expressed as
(47)$$\hat{B}=\left[\begin{array}{cc} {\hat{B}}_{1} & {\hat{B}}_{2} \end{array}\right]$$
where
(48)$${\hat{B}}_{i}=\left[\begin{array}{cccc} \frac{\partial N_{i}}{\partial x} & 0 & z\frac{\partial N_{i}}{\partial x} & \Phi^{k}\frac{\partial N_{i}}{\partial x}\\ 0 & R_{xz}^{k}{Μ}_{3}^{k}\frac{\partial N_{i}}{\partial x} & R_{xz}^{k}{Μ}_{1}^{k}N_{i} & R_{xz}^{k}{Μ}_{2}^{k}N_{i} \end{array}\right]$$
where $R_{xz}^{k}=1/Q_{44}^{k}$, ${Μ}_{j}^{k}$ are given in Equation (39), and *j* = 1~3.

By means of the following equation, the vector of nodal displacement $d^{e}$ can be expressed as
(49)$$K^{e}d^{e}=F^{e}+F_{p}^{e}$$
where $F^{e}$ and $F_{p}^{e}$ are the loading vector and the applied-electric-potential vector for an element, respectively.
(50)$$F^{e}=\int_{x_{i}}^{x_{i+1}}N^{T}Pdx$$
(51)$$F_{p}^{e}=\int_{x_{i}}^{x_{i+1}}\int_{-h/2}^{h/2+h_{p}}B^{T}\tilde{e}Edzdx$$
in which
(52)$$\tilde{e}=\left[\begin{array}{cc} 0 & e_{31}^{k}\\ S_{p}^{k}e_{15}^{k} & 0 \end{array}\right]$$
(53)$$S_{p}^{k}=\frac{2h_{p}}{e_{15}^{k}V_{t}\left(2z-h\right)}$$

Assembling all the element-stiffness matrices, the global equation is obtained in the following form:(54)$$Kd=F+F_{p}^{}$$
where K is the stiffness matrix; F and $F_{p}^{}$ are the global loading vector and the applied-electric-potential vector, respectively.

## 4. Numerical Results and Discussion

In this section, various numerical examples are presented in order to assess the proposed finite-element formulation. Additionally, the influences of the electro-mechanical loading, graphene distribution pattern, volume fraction, lamination scheme, thickness of the MFC-actuator layer, and length-to-thickness ratio on the static behaviors of FG-GRC-laminated beams with MFC actuators are carried out. 

### 4.1. Verification Study

As a typical example, a simply supported laminated beam (0°/90°/0°) subjected to a sinusoidal mechanical load $q=q_{0}\sin\left(\pi x/a\right)$ is firstly analyzed. The material characteristics of the laminated beam are [70]: $E_{1}=172.5\mathrm{GPa}$, $E_{1}/E_{2}=25$, $G_{12}=G_{13}=0.5E_{2}$, $G_{23}=0.2E_{2}$, $v_{12}=v_{13}=v_{23}=0.25$. Dimensionless results can be obtained by the relations: $$\bar{u}=\frac{E_{2}}{q_{0}h}u,\;\bar{w}=\frac{100E_{2}h^{3}}{q_{0}a^{4}},\;\left({\bar{\sigma }}_{x},\;{\bar{\tau }}_{xz}\right)=\frac{1}{q_{0}}\left(\sigma_{x},\;\tau_{xz}\right).$$

Convergence studies are first conducted, and the results with different mesh densities are presented in Table 1. It is found that the converged results agree well with the 3D elasticity solutions (Exact) [70]. In addition, the displacements and stresses converge at the mesh density of 48, which may be considered to be adequate for the thick laminated beam with a piezoelectric layer. Table 2 presents the displacements and stresses obtained from different models. The present results are compared with the exact solutions [70] and the results obtained from Reddy’s higher-order shear-deformation theory (HSDT-R), trigonometric higher-order shear-deformation theory (HSDT-S&G), first-order shear-deformation theory (FSDT) and classical lamination theory (CLT). The results of HSDT-R, HSDT-S&G, FSDT and CLT are provided by Sayyad and Ghugal [71]. In Table 2, it can be observed that the present model can produce more accurate results when compared with other models.

In order to further assess the present finite-element formulation, the simply supported FG-GRC-laminated beam with an MFC layer (0°/90°/0°/90°/0°/0°/90°/0°/90°/0°/M) that was subjected to a sinusoidal load $q=q_{0}\sin\left(\pi x/a\right)$ is investigated. The material properties for the matrix and graphene are [72]: $E^{m}=2.5\mathrm{GPa}$, $v^{m}=0.34$, $E_{22}^{G}=1807\mathrm{GPa}$, $E_{11}^{G}=1812\mathrm{GPa}$, $G_{12}^{G}=683\mathrm{GPa}$, $v_{12}^{G}=0.177$. The graphene efficiency parameters [33] for different volume fractions are shown in Table 3. The assumptions $G_{23}=G_{13}=0.5G_{12}$ are made for the numerical analysis [33]. The material constants for the MFC piezoelectric layer [50] are $E_{1}=30.336\mathrm{GPa}$, $E_{2}=E_{3}=15.857\mathrm{GPa}$, $G_{12}=G_{13}=G_{23}=5.515\mathrm{GPa}$, $v_{12}=v_{13}=0.31$, $v_{23}=0.438$, $d_{31}=-170\mathrm{pm}/V$, $d_{32}=-100\mathrm{pm}/V$. Dimensionless results can be obtained by the relation:$\left({\bar{\sigma }}_{x},\;{\bar{\tau }}_{xz}\right)=\left(\sigma_{x},\;\tau_{xz}\right)/q_{0}$. The thicknesses of FG-GRC-laminated beam and the MFC layer are h = 10 mm and hp = 0.5 mm, respectively. Figure 4, Figure 5, Figure 6 and Figure 7 present the distributions of the in-plane and transverse shear stresses in terms of thickness for the FG-GRC-laminated beams with MFC layers. The 3D elasticity solutions (Exact) in Figure 4, Figure 5, Figure 6 and Figure 7 were computed by the authors in terms of Pagano’s approach [70]. It can be seen from Figure 4, Figure 5, Figure 6 and Figure 7 that the present plate theory can accurately yield the stresses without using any post-processing procedure. However, Reddy’s higher-order theory (HSDT-R) [73], four-variable refined higher-order theories (HSDT-T [42], HSDT-A [40]) and first-order shear-deformation theory (FSDT) are apparently less accurate in predicting the transverse shear stresses. It is also observed that the HSDT-R, HSDT-T, HSDT-A and FSDT models are unable to satisfy the compatible conditions of transverse shear stresses. Therefore, for the interlaminar-stress analysis of FG-GRC-laminated beams with MFC layers, the compatible conditions of transverse shear stresses at the interfaces should be satisfied.

### 4.2. Parametric Study

In this section, parametric studies on the static analysis of FG-GRC-laminated beams with MFC actuators under electro-mechanical loads are conducted. The influences of the graphene volume fraction, distribution pattern, electro-mechanical loading, thickness of the MFC-actuator layer, lamination scheme, and the length-to-thickness ratio on the deformations and stresses of an FG-GRC-laminated beam with an MFC actuator under combined sinusoidal, mechanical and electrical loads are presented. Unless otherwise specified, the beams are simply supported, and the thicknesses of the FG-GRC-laminated beam and the MFC layer are *h* = 10 mm and *hp* = 0.5 mm, respectively. The lamination scheme is taken to be (0°/90°/0°/90°/0°/0°/90°/0°/90°/0°/M). The material characteristics of the matrix, graphene and MFC piezoelectric layer are the same as those used in the previous section. Dimensionless results can be obtained by the following relations:(55)$$\bar{u}=\frac{E_{m}}{q_{0}h}u,\;\bar{w}=\frac{100E_{m}h^{3}}{q_{0}a^{4}},\;\left({\bar{\sigma }}_{x},\;{\bar{\tau }}_{xz}\right)=\frac{1}{q_{0}}\left(\sigma_{x},\;\tau_{xz}\right)$$

Firstly, the FG-GRC-laminated beam with an MFC actuator under electro-mechanical loading (*q*_0_ = 100, V = 100) is investigated. Figure 8 depicts the normalized transverse deflections of the UD GRC plates with MFC-actuator layers. It is found that the transverse deflections are greatly influenced by the graphene volume fraction *V_G_*. The transverse deflections can be significantly reduced by adding a very small amount of graphene nanofillers. Therefore, graphene can help to improve the stiffness of the beam and suppress the deflections. 

Table 4 presents the transverse displacement and stresses of thick, laminated FG-GRC beams with MFC layers (*a/h* = 8) for the different graphene distributions and boundary conditions under a lateral load (*q*_0_ = 100). The following three boundary conditions (BCs) are considered: two ends of the beam are simply supported (SS), one end (*x* = 0) of the beam is clamped and the other end is simply supported (CS), and two ends of the beam are clamped (CC). In Table 4, it is found that the transverse displacement and stresses are strongly influenced by the boundary conditions. The more constraint that is applied to the ends of the beam, the more rigidity of the boundaries that is obtained. The results in Table 4 indicate that when other parameters are the same, the SS beam has the highest deflection and the CC beam has the lowest owing to the clamped end having more rigidity than the simply supported end.

The through-the-thickness variations of the in-plane displacements and stresses of the thick FG-GRC-laminated beam (*a/h* = 4) with an MFC actuator under electro-mechanical loading (*q*_0_ = 100, V = 100) for different graphene distribution patterns are shown in Figure 9. It can be seen from Figure 9 that the graphene distribution pattern has significant effects on the in-plane displacements and stresses of smart FG-GRC-laminated beams. Different graphene distribution patterns lead to different stiffnesses of the beam, which then lead to different displacements and stresses. Consequently, the deformations and stresses of smart FG-GRC-laminated beams can be regulated by changing the internal distribution pattern in engineering applications. In addition, it is also observed from Figure 9 that when a voltage is applied, the transverse shear stresses at the interface between the MFC piezoelectric layer and the FG-GRC beams will suddenly increase.

The normalized displacements and stresses of simply supported FG-GRC-laminated beams with MFC actuators under a sinusoidal mechanical load (*q*_0_ = 100) and electric loads (V = 0, V = 100 and V = −100) are shown in Table 5. The span-to-thickness ratio *a/h* is taken to be 10. In Table 5, it is found that the reversal of deflections and stresses can be observed due to the change in voltage polarity. In addition, the results in Table 5 reveal that the actuation responses of the displacements and stresses are very large. The effects of the thickness of the MFC-actuator layer on the actuation responses of a smart, thick FG-GRC-laminated beam (*a/h* = 5) subjected to an electro-mechanical loading (*q*_0_ = 100, V = 100) are shown in Table 6. The results in Table 6 indicate that the actuation effects on the deformations and stresses are decreased with the increase in the thickness of the MFC-actuator layer *h_p_*. Based on the above analysis, it can be concluded that the electro-mechanical loading and the activated MFC piezoelectric layer have remarkable impacts on the static analysis of FG-GRC-laminated beams with MFC actuators. 

Table 7 presents the effect of the length-to-thickness ratio (*a/h*) on the deflection and transverse shear stress of FG-X and FG-V GRC beams with an MFC actuator subjected to an electro-mechanical loading (*q*_0_ = 100, V = 100). It is found that the actuating response of the thick FG-GRC-laminated beams with MFC actuators is more significant than that of the thin beams. Additionally, it should be noted that the ${\bar{\tau }}_{xz}\left(h/2\right)$ is the transverse shear stress between the MFC piezoelectric layer and the FG-GRC beam. It can be seen from Table 7 that the actuation responses of the transverse shear stresses are decreased with the increase in *a/h*.

## 5. Conclusions

Based on an attractive electro-mechanical coupled-beam model, a C^0^-type finite-element formulation was developed for the interlaminar shear stress analysis of a thick FG-GRC-laminated beam with an MFC actuator. Using the HW variational principle, the improved transverse shear stresses in terms of the electro-mechanical coupling effect were derived. In order to assess the accuracy and efficiency of the present formulation, the results acquired from other models and the 3D elasticity solutions were also utilized for comparison. Furthermore, comprehensive parametric studies were carried out in order to analyze the effects of various parameters on the deformations and stresses. By analyzing typical static problems, some conclusions can be drawn:

(1) The present model can satisfy the compatible conditions of transverse shear stresses, and the results acquired from the present model agreed well with the 3D elasticity solutions.

(2) The stiffness of the FG-GRC-laminated beam with an MFC actuator can be significantly enhanced by adding a small amount of graphene. The displacements and stresses of the FG-GRC-laminated beam with an MFC layer are sensitive to the graphene distribution pattern. Therefore, the transverse shear stresses can be optimized by adjusting the graphene distribution pattern.

(3) The actuating responses of the thick FG-GRC-laminated beams with MFC actuators are more significant than that of the thin beams. When the electric voltages were applied, the maximum transverse shear stress was obtained at the interface between the MFC piezoelectric layer and the GRC laminates.

In addition, the effects of interlaminar stresses on the stability and dynamic behaviors of smart FG-GRC-laminated structures will be conducted in future work.

## Figures and Tables

**Figure 1 materials-14-07802-f001:**
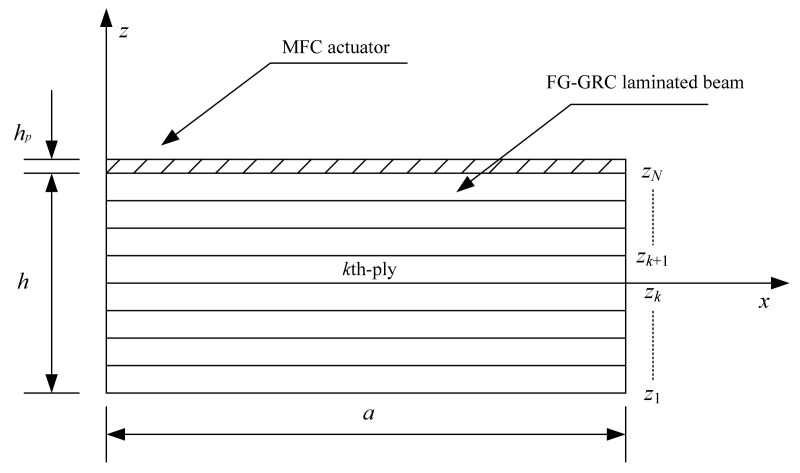
FG-GRC-laminated beam integrated with an MFC actuator.

**Figure 2 materials-14-07802-f002:**
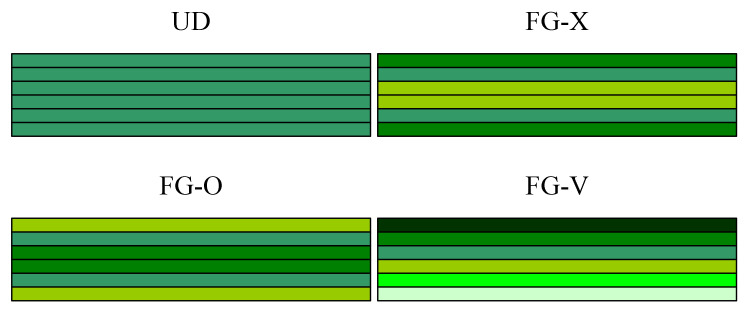
Configurations of FG-GRC-laminated beams with different distribution patterns.

**Figure 3 materials-14-07802-f003:**
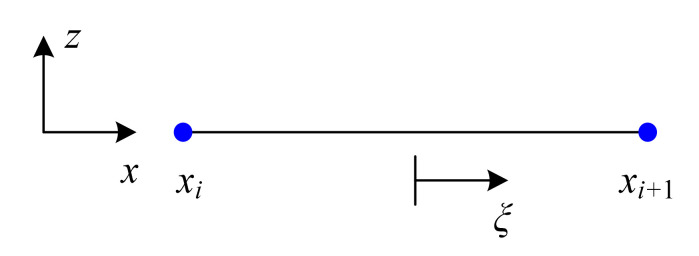
Two-node beam element.

**Figure 4 materials-14-07802-f004:**
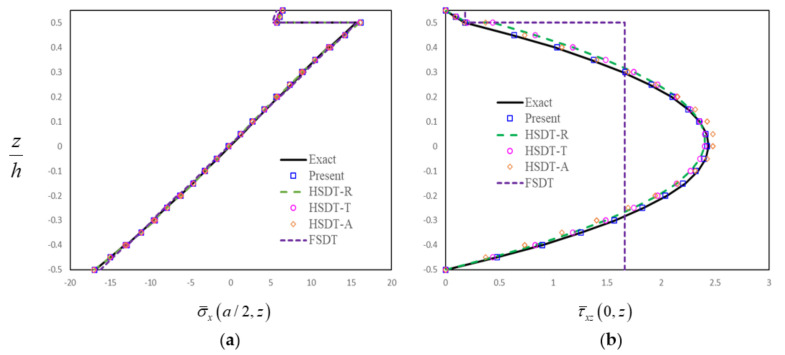
Distributions of stresses in terms of thickness for the FG-GRC beam with an MFC layer (*a/H* = 5, *H = h + h_p_*, UD): (**a**) Distributions of in-plane stress in terms of thickness; (**b**) Distributions of transverse shear stress in terms of thickness.

**Figure 5 materials-14-07802-f005:**
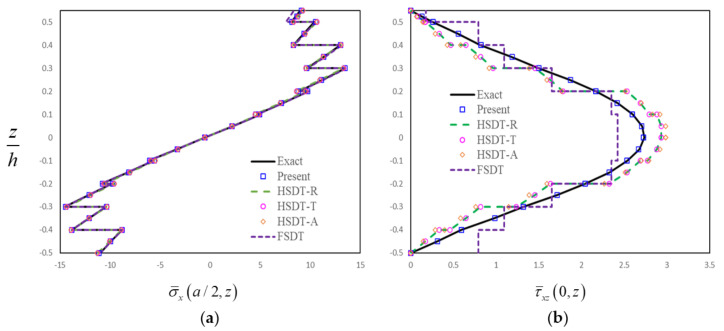
Distributions of stresses in terms of thickness for the FG-GRC beam with an MFC layer (*a/H* = 5, *H = h + h_p_*, FG-O): (**a**) Distributions of in-plane stress in terms of thickness; (**b**) Distributions of transverse shear stress in terms of thickness.

**Figure 6 materials-14-07802-f006:**
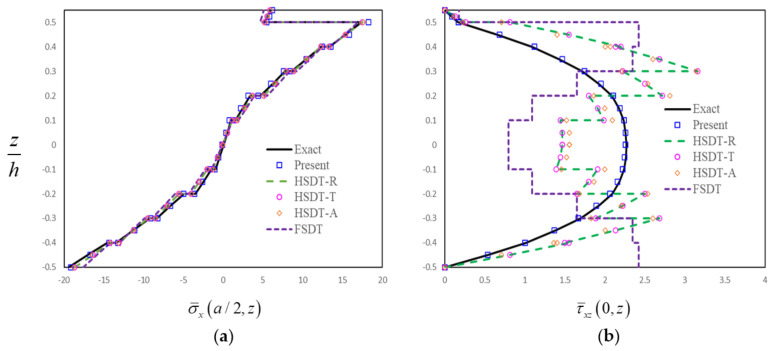
Distributions of stresses in terms of thickness for the FG-GRC beam with an MFC layer (*a/H* = 5, *H = h + h_p_*, FG-X): (**a**) Distributions of in-plane stress in terms of thickness; (**b**) Distributions of transverse shear stress in terms of thickness.

**Figure 7 materials-14-07802-f007:**
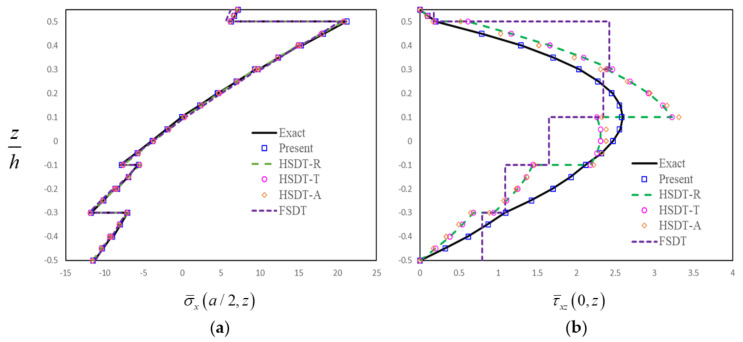
Distributions of stresses in terms of thickness for the FG-GRC beam with an MFC layer (*a/H* = 5, *H = h + h_p_*, FG-V): (**a**) Distributions of in-plane stress in terms of thickness; (**b**) Distributions of transverse shear stress in terms of thickness.

**Figure 8 materials-14-07802-f008:**
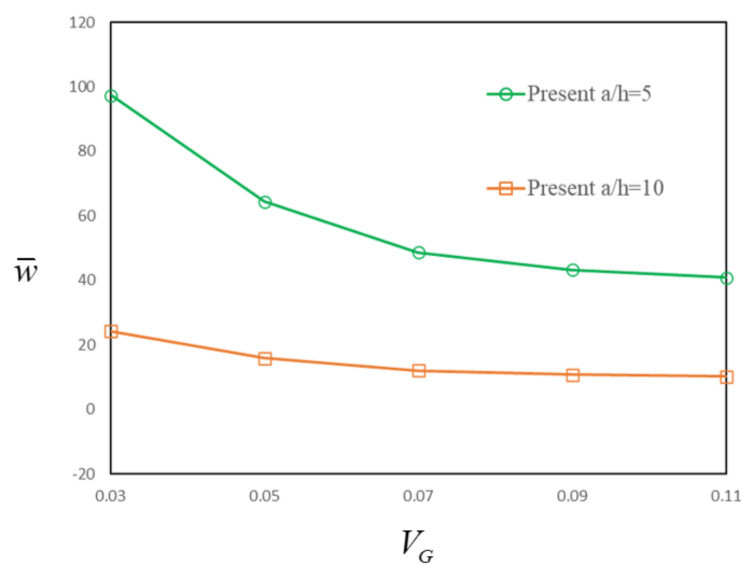
Transverse deflections for the FG-GRC beam with an MFC actuator for different graphene fraction volumes.

**Figure 9 materials-14-07802-f009:**
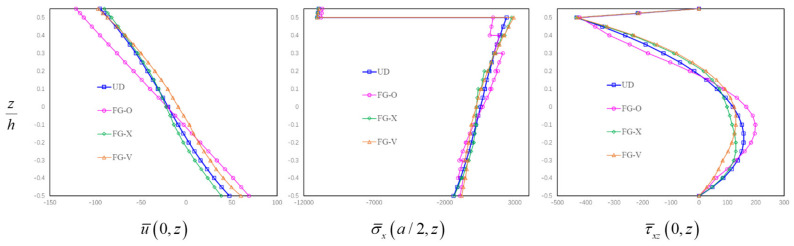
Variations of in-plane displacements and stresses through the thickness direction for the thick FG-GRC beams with an MFC actuator (*a/h* = 4).

**Table 1 materials-14-07802-t001:** Convergence of displacements and stresses for the cross-ply laminated plate (*a/h* = 4).

Mesh Density	Present (8)	Present (16)	Present (32)	Present (48)	Present (64)	Exact (70)
$\bar{w}\left(\frac{a}{2},0\right)$	2.8777	2.9033	2.9097	2.9109	2.9109	2.8870
$\bar{u}\left(0,-\frac{h}{2}\right)$	0.8876	0.8996	0.9026	0.9031	0.9033	0.9500
${\bar{\sigma }}_{x}\left(\frac{a}{2},-\frac{h}{2}\right)$	16.983	17.550	17.694	17.720	17.729	17.950
${\bar{\tau }}_{xz}\left(0,0\right)$	1.3866	1.4210	1.4297	1.4314	1.4319	1.4300

Note: The numbers in bracket denote the number of elements.

**Table 2 materials-14-07802-t002:** Comparison of displacements and stresses obtained from different models (*a/h* = 4).

Models	$\bar{w}\left(\frac{a}{2},0\right)$	$\bar{u}\left(0,-\frac{h}{2}\right)$	${\bar{\sigma }}_{x}\left(\frac{a}{2},-\frac{h}{2}\right)$	${\bar{\tau }}_{xz}\left(0,0\right)$
Exact [70]	2.8870	0.9500	17.950	1.4300
Present	2.9109	0.9033	17.729	1.4319
HSDT-R [71]	2.6985	0.8640	17.007	1.5565
HSDT-S&G [71]	2.7342	0.8885	17.575	1.5278
FSDT [71]	2.4094	0.5124	10.085	1.7690
CLT [71]	0.5097	0.5124	10.085	1.7690

**Table 3 materials-14-07802-t003:** The efficiency parameters with respect to different volume fractions of graphene.

$V_{G}$	$\eta_{1}$	$\eta_{2}$	$\eta_{3}$
0.03	2.929	2.855	11.842
0.05	3.068	2.962	15.944
0.07	3.013	2.966	23.575
0.09	2.647	2.609	32.816
0.11	2.311	2.260	33.125

**Table 4 materials-14-07802-t004:** Normalized displacements and stresses of FG-GRC-laminated beams (*a/h* = 8) with MFC layers.

BCs		$\bar{w}\left(\frac{a}{2},0\right)$	${\bar{\sigma }}_{x}\left(\frac{a}{2},\frac{h}{2}\right)$	${\bar{\tau }}_{xz}\left(0,0\right)$
SS	UD	0.3657	36.851	3.7214
	FG-X	0.3514	40.603	3.4998
	FG-V	0.4632	47.908	3.9528
CS	UD	0.1794	19.809	3.5334
	FG-X	0.1845	22.534	1.5760
	FG-V	0.2197	25.690	3.8135
CC	UD	0.1086	13.707	2.6912
	FG-X	0.1167	15.627	1.2096
	FG-V	0.1297	17.973	2.8854

**Table 5 materials-14-07802-t005:** Normalized transverse displacements and stresses of FG-GRC-laminated beams (*a/h* = 10) integrated with a PFRC actuator.

Distribution Pattern	Volt	$\bar{w}\left(\frac{a}{2},0\right)$	$\bar{u}\left(0,\frac{h}{2}\right)$	${\bar{\sigma }}_{x}\left(\frac{a}{2},\frac{h}{2}\right)$	${\bar{\tau }}_{xz}\left(0,\frac{h}{2}\right)$
UD	0	0.3516	−5.0846	57.260	0.3378
	100	10.366	−211.23	2379.1	−172.99
	−100	−9.6628	201.06	−2264.3	173.66
FG-O	0	0.5012	−7.3808	38.051	0.4898
	100	15.032	−287.90	1484.6	−167.95
	−100	−14.030	273.13	−1408.2	168.93
FG-X	0	0.3311	−4.6454	62.620	0.3092
	100	9.4909	−201.25	2713.5	−173.67
	−100	−8.8286	191.96	−2588.8	174.28
FG-V	0	0.4488	−5.5146	74.337	0.3698
	100	11.402	−205.03	2764.7	−173.31
	−100	−10.505	194.01	−2615.4	174.05

**Table 6 materials-14-07802-t006:** Effects of the thickness of MFC-actuator layer on the actuation responses of FG-GRC-laminated beam (*a/h* = 5) integrated with an MFC actuator.

Graphene Distribution Pattern	*h_p_*	$\bar{w}\left(\frac{a}{2},0\right)$	$\bar{u}\left(0,\frac{h}{2}\right)$	${\bar{\sigma }}_{x}\left(\frac{a}{2},\frac{h}{2}\right)$	${\bar{\tau }}_{xz}\left(0,\frac{h}{2}\right)$
UD	0.5 mm	42.002	−105.94	2386.8	−345.73
	1 mm	40.593	−100.82	2271.0	−318.12
	1.5 mm	38.867	−95.321	2146.9	−291.64
FG-X	0.5 mm	39.930	−101.30	2731.8	−347.04
	1 mm	38.947	−97.315	2623.6	−320.29
	1.5 mm	37.594	−92.883	2504.1	−293.94

**Table 7 materials-14-07802-t007:** Effects of length-to-thickness ratio (a/h) on the actuation responses of FG-GRC-laminated beam integrated with an MFC actuator.

Graphene Distribution Pattern	*a/h*	$\bar{w}\left(\frac{a}{2},0\right)$	${\bar{\tau }}_{xz}\left(0,\frac{h}{2}\right)$
FG-X	5	39.930	−347.04
	10	9.4909	−173.67
	20	2.5478	−86.433
FG-V	5	46.340	−345.73
	10	11.402	−173.31
	20	3.1359	−86.204

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
