# Peer review of "An Alternative Electro-Mechanical Finite Formulation for Functionally Graded Graphene-Reinforced Composite Beams with Macro-Fiber Composite Actuator"

_materials, 2021, doi:10.3390/ma14247802_

Round 1

Reviewer 1 Report

Fu et al have presented a theoretical/numerical study on the mechanical properties of a beam composed by functionally graded graphene reinforced composite (FG-GRC) laminated thick beam integrated with piezoelectric macro fiber composite (MFC) actuators under electro-mechanical loadings. Generally, they present a finite element formulation for taking into account interlaminar shear stress analysis, which is not suitable considered by previous models existing in the literature. The authors keep control of the accuracy of their model when by comparing with other previously available from the literature. They also calculate several parametric studies by varying deformation and stresses. They found the stiffness and the transverse shear stresses can be improved by varying the amount of graphene, and that laminated beams are more sensitive to the MFC actuator as compared to thin beams.

The study is clearly written, and it should be in the interest of many scientists working in material sciences. In principle, the authors provide a reliable model which could be used for further simulations for constructing such devices. As such, I can endorse publication in MDPI’s Materials ate the authors consider the following comments,

  • In general the use of English language is good, but the manuscript needs to be improved. There are many word glued together (“conversionefficiency” line 45, “andgraphene” line 60, “theeffectiveelastic” line 168, “continuityconditions” line 201, etc many more). Some typos (“as shown” should read “are shown” line 156, “It is can” lines 377,439). Some grammatical sentences (lines 401-402)
  • Figures 4-11 need to improve their quality. When one zooms in, they become very blurry. Also, increase the size of the labels as they are barely seeable. In Figure 2, please explain in the caption what each color means.
  • Right after Eq. 4, define what u and w on the left-hand-side of the equations are. Right after Eq. 8, define what Q44 is. Eqs. 50-53 seem to have a problem with the power of B, please fix it.
  • In Table 1, please indicate in the caption what each number inside the parenthesis means.
  • How do the authors explain that in Fig. 9, FG-O behaves linearly as compared to the other three? Please discuss.
  • What is the acronym “PFRC” on line 464? Please, define it.
  • From an electrostatic point of view, graphene is a semimetal; from a mechanical point of view, graphene is strong. How are the authors’ results connected with these simple facts? (Specifically Points (2) and (3) of the Conclusions). Please discuss.

Reviewer 2 Report

Dear Authors

The manuscript presented concerns an interesting and actual subject. This manuscript can be accepted after major revision. The following suggestion and comments should be taken:

  1. The overall English needs to be improved. Please seek guidance from a native English speaker if possible (commas, plural form,  "the" "a", and others could be corrected).
  2. (Line 60) Please correct "and graphene" - please insert a space.
  3. (Line 60-62) Please add more articles about such composites and enhancement their potential applications. Please cite (1) Nanomaterials 2021, 11(8), 2080; https://doi.org/10.3390/nano11082080, (2) Materials 2021, 14(9), 2448; https://doi.org/10.3390/ma14092448 , (3) J. Compos. Sci. 2021, 5(9), 234; https://doi.org/10.3390/jcs5090234.
  4. Figure 4 please correct this image for better quality (the inscriptions on the drawing).
  5. Figures 5,6, 7, 8, 9, 10 and 11 are the same problem. Please correct this image for better quality (the inscriptions on the drawing).
  6. Add more information about potential deformations. What about carbon ends saturated with hydrogen atoms in carbon materials? Please explain in the comments. The dangling bonds of carbon ends may affect the structural and electronic properties of edge states. 
  7. Line 510 this is patent or grant? If grant please change it to "Acknowledgements"
  8. Could the authors include the standard deviation of the methods?
  9. Why authors choose graphene for the study? Please explain.
  10. Authors are suggested to describe some future plans in conclusions to enhancement

Reviewer 3 Report

This research is about modeling for FG-GRC laminated thick beam with MFC actuator. It is more about modeling than materials. I doubt if the topic is suitable for the Materials special issue “Advanced Composite Material Design and Manufacturing Technology for Aerospace Engineering”. Some minor issues are also listed below:

  1. The language should be improved. For example, “Four distribution patterns of FG-GRC laminated beams, namely UD, FG-V, FG-O and FG-X, as shown in Fig. 2” (section 2.1) is not a complete sentence.
  2. In the experimental section, more explanation is necessary for Figure 2. What is the meaning for each color? And what are the meanings for UD, FG-V, FG-O and FG-X?
  3. Some figures can be merged (like Figures 9, 10, 11 can be merged into panels of one figure).
  4. What is the role of graphene here? And what are the mechanisms behind?

Round 2

Reviewer 2 Report

Accept in present form

Reviewer 3 Report

I still suggest authors add more information on graphene to the background section. Others are fine.